# The Heterogeneous Impact of Prediagnostic Folate Intake for Fluorouracil-Containing Induction Chemotherapy for Head and Neck Cancer

**DOI:** 10.3390/cancers15215150

**Published:** 2023-10-26

**Authors:** Michi Sawabe, Daisuke Kawakita, Isao Oze, Shinichi Iwasaki, Yasuhisa Hasegawa, Shingo Murakami, Hidemi Ito, Nobuhiro Hanai, Keitaro Matsuo

**Affiliations:** 1Division of Cancer Epidemiology and Prevention, Department of Preventive Medicine, Aichi Cancer Center Research Institute, 1-1 Kanokoden, Chikusa-ku, Nagoya 464-8681, Japan; michi731@hotmail.co.jp (M.S.); i_oze@aichi-cc.jp (I.O.); kmatsuo@aichi-cc.jp (K.M.); 2Department of Head and Neck Surgery, Aichi Cancer Center Hospital, 1-1 Kanokoden, Chikusa-ku, Nagoya 464-8681, Japan; hanai@aichi-cc.jp; 3Department of Otolaryngology, Head and Neck Surgery, Nagoya City University Graduate School of Medicine, 1, Kawasumi, Mizuho-cho, Mizuho-ku, Nagoya 467-8601, Japan; iwasaki0824@gmail.com; 4Department of Head and Neck Surgery, Asahi University Hospital, 3-23 Hashimoto-cho, Gifu 500-8523, Japan; yhasegawa@hosp.asahi-u.ac.jp; 5Department of Otolaryngology, Head and Neck Surgery, Nagoya City East Medical Center, 1-2-23, Wakamizu, Mizuho-ku, Nagoya 464-8547, Japan; murakamishingo1953@gmail.com; 6Division of Cancer Information and Control, Department of Preventive Medicine, Aichi Cancer Center Research Institute, 1-1 Kanokoden, Chikusa-ku, Nagoya 464-8681, Japan; hidemi@aichi-cc.jp

**Keywords:** folate intake, head and neck squamous cell carcinoma, induction chemotherapy, fluorouracil

## Abstract

**Simple Summary:**

To our knowledge, this is the first study to suggest that an association between prediagnostic folate intake and head and neck squamous cell carcinoma (HNSCC) survival significantly differs based on fluorouracil (FU)-containing chemotherapy. In the FU-containing chemotherapy group, a higher folate intake was significantly associated with better overall survival; however, no apparent association between prediagnostic folate intake and survival was observed in definitive treatment without an FU-containing chemotherapy group. Fluorouracil exerts its antitumor activity by inhibiting folate-mediated one-carbon metabolism; therefore, our finding indicates that in the carcinogenic process, the folate status causes HNSCC to be heterogeneous in terms of one-carbon metabolism.

**Abstract:**

Fluorouracil (FU) exerts its antitumor activity by inhibiting folate-mediated one-carbon metabolism. Evidence that folate may play a role in the carcinogenic process via folate-mediated one-carbon metabolism has given rise to the hypothesis that pre-diagnostic folate intake may induce heterogeneous chemosensitivity to FU-containing induction chemotherapy (IC) in head and neck cancer. To assess this hypothesis, we conducted a cohort study to investigate whether the association between prediagnostic dietary folate intake and cancer survival differed between treatment regimens with and without FU-containing IC in 504 cases of locally advanced (stage III/IV) HNSCC, using an epidemiologic database combined with clinical data. In total, 240 patients were treated with FU-containing IC followed by definitive treatment, and 264 patients were treated with definitive treatment alone. Definitive treatment is defined as (1) the surgical excision of a tumor with clear margins, with or without neck lymph node dissection; or (2) radiotherapy with or without chemotherapy. In the overall cohort of the FU-containing IC group, a higher folate intake was significantly associated with better overall survival (adjusted hazard ratios (HRs) for the highest compared to the lowest folate tertiles (HR_T3-T1_) = 0.42, 95%CI, 0.25–0.76, P_trend_ = 0.003). Conversely, no apparent association between prediagnostic folate intake and survival was observed with definitive treatment alone (HR_T3-T1_: 0.83, 95%CI, 0.49–1.42, P_trend_ = 0.491)). A consideration of the cumulative dose of FU-containing IC showed that the survival impact of prediagnostic folate intake differed statistically significantly by treatment regimen (Pinteraction = 0.012). In conclusion, an association between prediagnostic folate intake and HNSCC survival significantly differed by FU-containing IC. This finding indicates that in the carcinogenic process, folate status causes HNSCC to be heterogenous in terms of one-carbon metabolism.

## 1. Introduction

Squamous cell carcinoma of the head and neck (HNSCC) represents a substantial global health challenge, with over 700,000 new diagnoses each year [1]. Most patients diagnosed with HNSCC exhibit locally advanced or locoregionally advanced stages of the disease [2]. Despite comprehensive treatments, including radical surgical resection or combined radiotherapy and chemoradiotherapy, overall cure rates have not been optimal [3]. Induction chemotherapy offers potential advantages, such as addressing micro-metastases and promoting the preservation of head and neck organs [4]. The use of fluorouracil (FU)-containing induction chemotherapy, preceding definitive treatment, has been recognized as a potent therapeutic option for patients with locally advanced HNSCC; however, identifying the appropriate patient subgroup for this regimen remains a point of discussion [5].

Folate-mediated one-carbon metabolism is essential to the synthesis of DNA and the methylation and regulation of chromatin structure. Fluorouracil (FU), one of the most widely used anticancer drugs for many cancers, including head and neck, exerts its effect via the inhibition of this folate-mediated one-carbon metabolism [6]. Once metabolized, FU becomes 5-fluorodUMP, interacting with thymidylate synthase (TS) and 5–10-methylene tetrahydrofolate to establish a tripartite complex (as shown in Figure 1 [6,7]). This interaction continuously inhibits TS, consequently impeding DNA synthesis, which is believed to be the primary anticancer action of FU.

The dependence of the antitumor activity of FU on folate-mediated one-carbon metabolism explains the important role of folate, a water-soluble compound also known as vitamin B9, in the antitumor effect of FU [6]. It has been reported that the combination of folate and FU may produce tighter ternary complexes and improve the antitumor efficacy of FU [6,8,9]. Indeed, the combination of these two is a critical component in the standard treatment of gastrointestinal cancers [10,11,12,13,14,15,16]. However, evidence for the impact of an association between folate intake and FU on clinical outcomes is limited.

Results from several experiments suggest that folate may play a role in the carcinogenic process via folate-mediated one-carbon metabolism due to the function of folate in nucleotide synthesis; the provision of folate to early tumoral lesions once they are established appears to increase tumorigenesis [17,18,19,20]. These results have given rise to the hypothesis that the heterogenous nutritional condition of folate in the carcinogenic process—in brief, pre-diagnostic folate intake—may induce tumors with heterogeneous chemosensitivity to FU’s inhibition of folate-mediated one-carbon metabolism. To date, however, only one study has provided epidemiologic evidence for an association of dietary folate intake prior to diagnosis with the efficacy of FU. Shitara et al. suggested a survival benefit for prediagnostic folate intake among patients with advanced gastric cancer treated with FU-based chemotherapy [21]. Our previous study evaluated the survival benefit of folate intake among patients with head and neck squamous cell carcinoma (HNSCC) overall; however, the question of whether an association between folate intake and survival differs by treatment, particularly FU-containing chemotherapy, remains unanswered. In response, we focused on the fact that FU is an essential component of induction chemotherapy (IC) for locally advanced HNSCC [22,23,24]. A comparison of the survival impact of prediagnostic folate intake between an FU-containing IC group (FU-containing IC followed by definitive treatment) and non-IC group (definitive treatment alone) would make it possible to assess whether chemosensitivity to FU-containing chemotherapy is heterogenous by folate status in the carcinogenic process.

Here, we report a cohort study designed to investigate whether the association of pre-diagnostic dietary folate intake and survival differed between IC including or not including FU among patients with locally advanced HNSCC in a Japanese population, using an epidemiologic database combined with clinical data.

## 2. Materials and Methods

The aim of this hospital-based cohort study was to specifically evaluate whether the association between prediagnostic dietary folate intake and survival differed between patients with locally advanced HNSCC treated with FU-containing IC followed by definitive treatment and with definitive treatment alone.

This paper sourced its patient data from the Hospital-based Epidemiologic Research Program at Aichi Cancer Center (HERPACC) which was carried out in two versions, with version 2 spanning from 2001 to 2005 and version 3 conducted from 2005 to 2012. The follow-up continued until January 2017. The HERPACC initiative, as documented earlier [25,26], involves new outpatients at the Aichi Cancer Center Hospital (ACCH), detailing each patient’s lifestyle prior to the onset of symptoms. All participants provided written informed consent to participate. In HERPACC versions 2 and 3, 29,736 and 28,337 first-visit outpatients were eligible, respectively, and of those who agreed to participate, 28,776 (97.1%) and 18,849 (66.4%) completed the questionnaire satisfactorily and enrolled in HERPACC. In the present HERPACC study, responses were systematically collected and checked by trained interviewers. The information was then integrated into the HERPACC system and updated routinely for cancer occurrence and survival status. The Aichi Cancer Center’s Institutional Ethics Committee approved this study (approval numbers: ACC-2022-0-341 and ACC-2022-0-302).

In the present analysis, we focused on patients registered in HERPACC who were diagnosed with primary head and neck cancers based on the following specifics: (1) primary head and neck cancers located in the oral cavity, oropharynx, hypopharynx, or larynx; (2) a confirmed squamous cell carcinoma diagnosis; (3) locoregionally advanced cancer (stage III/IV) and no history of recurrence or metastasis; (4) receiving definitive therapy with or without FU-containing IC at the Aichi Cancer Center (ACC); and (5) PS ≤ 2, according to the Eastern Cooperative Oncology Group criteria. Eligibility spanned cancers identified by the given ICD10 codes, capturing cancers from regions including the oral cavity, oropharynx, hypopharynx, and larynx.

For the present study, 1013 participants with head and neck cancer were available as a baseline population. Of these, 509 patients were excluded: 108 patients were excluded due to recurrent disease or metastasis, 46 were excluded due to a diagnosis of not squamous cell carcinoma (SCC), 345 were excluded in stages 1–2, and 10 were excluded due to missing data on folate intake, as shown in Appendix A. Finally, the analysis included 504 patients with locally advanced HNSCC, consisting of 240 patients treated with FU-containing IC followed by definitive treatment and 264 patients treated with definitive treatment alone.

### 2.1. Estimation of Folate Intake and Life Style Factors

Data on environmental risk determinants were obtained through a participant-completed survey. This survey prompted participants to reflect on their exposure to risk factors prior to noticing the symptoms that led them to seek hospital care. Professionally trained personnel reviewed the completed questionnaires for accuracy.

The HERPACC survey was the basis for estimating folate consumption, encompassing queries about demographics, personal and familial medical backgrounds, physical metrics, activity levels, tobacco and alcohol consumption, vitamin supplementation, and specific food and drink intake. A Food Frequency Questionnaire (FFQ) [27,28], comprising 47 individual food entities categorized under eight frequency options, was utilized to map out dietary habits. To determine the average nutrient consumption per day, we multiplied the quantity of food (in grams) or portion size by its nutrient density per 100 g, referencing the Standard Tables of Food Composition in Japan, Fifth Edition [29]. For primary foods like rice, noodles, and bread, portion specifics were sought. Nutrient intake from supplements was not incorporated into the total vitamin count due to the non-specific nature of the multivitamin questionnaire [28]. Given the limited usage of folic acid supplements in Japan, we nonetheless factored in vitamin supplement usage in our multivariate analysis to ensure accuracy. We also performed a supplementary analysis on non-supplement users to exclude potential supplementation effects. The validity and consistency of the FFQ were established by comparing it to a standard 3-day weighted dietary log (3d-WDRs) [27,28]. After adjusting for energy intake, folate intake values were processed using a known residual method [30]. De-attenuated, log-transformed, and energy-adjusted Pearson’s correlation coefficients between dietary folate intake quantified with the FFQ and 3d-WDRs were 0.36 (95% confidence interval (95%CI): 0.12–0.58) in men and 0.38 (95%CI, 0.25–0.62) in women, respectively [28]. For this research, participants were classified into three tiers based on their folate intake levels over the research study’s duration. Smoking habits were assessed in pack-years (PYs) and categorized into never, light (PYs < 20), moderate (PYs < 40), and heavy smokers (PYs ≥ 40). Alcohol intake was translated into daily ethanol grams, segmenting participants into non-drinkers, light drinkers (<23 g ethanol/day), moderate drinkers (23–46 g ethanol/day), and heavy drinkers (>46 g ethanol/day).

### 2.2. Evaluation of Treatment

Details of the ACC’s treatment strategy were described previously [31,32,33,34,35]. In brief, all patients included in this analysis received definitive treatment with or without FU-containing IC prior to definitive treatment. Definitive treatment consisted of either surgery or radiotherapy (RT; 66–70 Gy) with or without platinum-based chemotherapy [34]. FU-containing IC was considered primarily for the purpose of organ preservation or improved survival outcome in patients with advanced stage (Union for International Cancer Control (UICC) stage III, IV) and progressive disease with lymph metastasis or tumor invasion [31,34]. At the ACC, the basic regimen of FU-containing IC was two cycles of a combination of FU and cisplatin every three weeks (tri- weekly FP, Appendix A) [31,32,33,34,35,36]. Regimens other than those with FP contained either FU or S-1, which is a prodrug of FU (Appendix A). In the present study, the cumulative dose of FU during IC was estimated using the regimen and number of cycles performed. Patients were then divided into two groups, those with a high cumulative dose of FU during IC (high-dose FU-containing IC, FU more than 8000 mg/m^2^, equivalent to two cycles of tri-weekly FP) and those with a low cumulative dose of FU (low-dose FU-containing IC, FU less than 8000 mg/m^2^).

Definitive treatment was performed within three weeks after IC in accordance with Response Evaluation Criteria in Solid Tumors (RECIST); patients diagnosed with a complete response or partial response were treated via RT, while others underwent surgery [37]. Salvage surgery was considered for residual disease after radiological treatment [37,38]. Postoperative RT was carried out for patients with a positive surgical margin or extracapsular spread to lymph nodes after surgery [37,38]. Patient characteristics and clinical features were recorded, including disease history, physical examination, performance status, imaging, and laboratory examination. Tumor staging was performed according to the Union for International Cancer Control (UICC) TNM Classification [39]. Patient evaluations and decision making were conducted by a conference of head and neck surgeons, a radiation oncologist, and reconstructive surgeons.

### 2.3. Statistical Analysis

In our analysis, we primarily examined dietary folate consumption, categorized into tertiles, to assess its relationship with several clinical outcomes. The study’s principal outcome was overall survival (OS), defined as the time from HNSCC diagnosis to either any-cause mortality or the most recent follow-up, whichever came first. Our secondary outcomes encompassed recurrence-free survival (RFS)—the duration from HNSCC diagnosis to either loco-regional or metastatic recurrence, any-cause death, or the most recent follow-up—and distant metastasis-free survival (DMFS), which is the time elapsed from diagnosis to metastatic recurrence, any-cause death, or the last recorded follow-up. We ascertained parameters such as patient survival, disease trajectory, and treatment regimen by perusing medical documentation. For patients unaccounted for in the follow-up, we validated their survival status via the annual census records.

The OS, RFS, and DMFS were estimated by the Kaplan–Meier method [40], and differences were evaluated using the log-rank test. To assess the association between folate intake and survival, we estimated multivariable hazard ratios (HRs) and 95% confidence intervals (CIs) using Cox proportional hazard models. The proportional hazards (PHs) assumption was evaluated graphically (log–log plot) and using a test for PH based on Schoenfeld residuals. A stratification analysis was performed by the cumulative dose of FU during IC (high- and low-cumulative doses of FU during FU-containing IC), definitive treatment (surgery or radiotherapy/chemoradiotherapy), and other clinical confounders (sex, age, ECOG PS, smoking, alcohol consumption, primary site, UICC T, and N classification or UICC stage). In addition to folate intake, we considered the following confounders in multivariate analyses: sex (male and female), age (continuous), Eastern Cooperative Oncology Group performance status (ECOG PS) (0, 1, and 2), cumulative smoking (non, light, moderate, and heavy), alcohol consumption (non, light, moderate, and heavy), primary site (oral cavity, oropharynx, hypopharynx, and larynx), UICC T classification (1, 2, 3, and 4), N classification (0, 1, 2, and 3) or UICC stage (III, IV), definitive treatment (surgery or radiotherapy), energy (continuous), supplemental vitamin use (yes or no), and study period (HERPACC 2 and 3, treated as strata). A sensitivity analysis was performed among the patients without supplement use. Interactions between folate intake and FU-containing IC were examined in a multivariable Cox regression model which included an interaction term between dietary folate intake and FU-containing IC. We defined an interaction term between folate consumption and FU-containing IC as a product of folate in tertiles (0/1/2) and an indicator variable of FU-containing IC (0: definitive treatment; 1: FU-containing IC). In addition, a lead-time bias by IC may overestimate the association of folate intake on survival in FU-containing IC. To reduce the lead-time bias, a landmark analysis was performed, with two cycles of tri-weekly FP required in a period of two months [31,34].

The distribution of patient demographics was evaluated using the chi-squared test or Fisher’s exact test, depending on suitability. We performed all computations utilizing Stata SE, version 16 (StataCorp, College Station, TX, USA). A threshold of *p* < 0.05 was set to denote statistical significance.

## 3. Results

### 3.1. Patient Characteristics

The median follow-up at the time of analysis was 60 months (range: 4–112). The five-year OS values for the FU-containing IC followed by the definitive treatment group and the definitive treatment alone group were 59.4% (95%CI: 52.5–65.7) and 62.0% (95%CI: 55.2–68.1), respectively, showing no significant difference (adjusted HR for the FU containing-IC followed by definitive treatment group compared to the definitive treatment alone group: 1.01, 95%CI 0.71–1.41, *p* = 0.976). The baseline demographic characteristics of the locally advanced HNSCC patients are shown in Table 1. The majority of patients had no supplement use (84%). Patients with oropharyngeal and hypopharyngeal cancer, stage 4, N2 disease, those receiving radiotherapy as definitive treatment, and heavy drinkers and smokers were more prevalent in the FU-containing IC followed by definitive treatment group compared with the definitive treatment alone group. The majority of patients in the FU-containing IC group received tri-weekly FP (89%, Appendix A). Patients with oral cavity cancer, UICC stage, those who were treated via surgery, and heavy drinkers were more prevalent in HERPACC version 3 than in version 2 (Appendix A).

### 3.2. Impact of Dietary Folate Intake on Overall Survival

In the FU-containing IC group overall, a higher folate intake was significantly associated with better survival, as shown in Table 2. Compared to those with low folate intake, patients with high intake showed better overall survival (5-year OS: 70.8 (59.0–80.6) vs. 50.8 (37.6–61.6), respectively; log-rank *p* =0.020, crude HR_T3-T1_: 0.50 (95%CI, 0.30–0.84), P_trend_ = 0.009, Figure 2A; upper part of Table 2). Even after adjusting for confounders, patients with high folate intake had significantly lower HRs for death than those with low folate intake, with an adjusted HR of 0.42 (95%CI, 0.25–0.76, P_trend_ = 0.004). Further, when stratified by the cumulative dose of FU during IC, among the patients treated with high-dose FU-containing IC, a 78% reduction in death was observed when comparing high and low folate intakes (adjusted HR_T1-3_ of 0.22 (95%CI: 0.10–0.53), P_trend_ < 0.001; middle part of Table 2, Upper section of Figure 3).

This association was similarly consistent across treatment modalities, even after stratification by the cumulative dose of FU (surgery group: overall FU containing-IC HR_T3-T1_ 0.35; 95%CI 0.13–0.93, P_trend_ = 0.034, high-dose adjusted HR_T3-T1_ 0.03, 95%CI 0.00–0.38, P_trend_ = 0.005; radiotherapy group: overall adjusted HR_T3-T1_ 0.56; 95%CI 0.26–1.17, P_trend_ = 0.121, high-dose adjusted HR_T3-T1_ 0.30; 95%CI; 0.10–0.91, P_trend_ = 0.033, left section, middle and bottom part of Table 2).

In contrast, among patients in the definitive treatment alone group, an association between folate intake and survival was not obvious, with an adjusted HR for high compared to low folate intake of 0.83 (95%CI, 0.49–1.42, P_trend_ = 0.491, 5-year OS: 67.6 (55.7–77.0) vs. 58.6 (45.8–69.2), log-rank *p* =0.447; Figure 2B upper part, right section of Table 2). This association was retained after stratification by treatment (surgery group: adjusted HR_T3-T1_ 0.89, 95%CI 0.33–2.38, P_trend_ = 0.823; radiotherapy group: adjusted HR_T3-T1_ 0.70, 95%CI 0.35–1.42, P_trend_ = 0.319, right section, middle-bottom part of Table 2).

In a comparison of the FU-containing IC group overall and the definitive treatment alone group, the difference in survival impact by folate intake and FU-containing IC did not reach statistical significance (P_interaction_ = 0.202; Table 2, upper section of Figure 3). However, when considering the cumulative dose of FU, the association of folate intake and overall mortality did statistically significantly differ between the high-dose FU-containing IC and definitive alone groups (P_interaction_ = 0.012; Table 2, upper section of Figure 3).

#### 3.2.1. Sensitivity Analysis and Stratification by Clinical Confounder

A sensitivity analysis performed in patients without supplement use showed heterogeneity more clearly. A similar trend was seen in the impact of pre-diagnostic folate on survival, and the heterogeneity of the survival impact of folate by FU-containing chemotherapy overall was potentially significant (*p* = 0.077, lower section of Figure 3). Moreover, this trend of an association between prediagnostic folate intake and survival by FU-containing IC was consistently observed in both HERPACC version 2 and version 3 (Appendix A). Furthermore, a landmark analysis showed consistent results (Appendix A).

When stratified by clinical confounders, the FU-containing IC group showed an overall survival benefit of folate intake across all subgroups other than the early T1-2 classification (Appendix A). Conversely, in the definitive treatment alone group, the association between folate intake and survival was absent across all subgroups other than female and laryngeal cancer (Appendix A).

#### 3.2.2. Impact of Dietary Folate Intake on Recurrent-Free Survival and Distant Metastasis-Free Survival

Folate intake showed no apparent association with RFS in either the FU-containing IC group or the definitive treatment alone group, with adjusted HRs for high compared to low folate intake of 0.88 (95%CI, 0.58–1.36, P_trend_ = 0.596, 5-year-RFS 44.4 (33.0–55.0) vs. 41.2 (30.1–52.2), log-rank *p* = 0.609; Figure 4A, upper section of Appendix A) and 0.82 (95%CI, 0.51–1.30, P_trend_ = 0.395, 5-year-RFS 57.0 (45.0–67.3) vs. 40.4 (28.4–52.3) log-rank *p* = 0.351, Figure 4B), respectively. This lack of association between folate intake and RFS was consistent after stratification by cumulative dose of FU.

In contrast, similar to OS, a comparison of high and low folate intakes showed that patients with a higher intake in the FU-containing IC group had better distant metastasis-free survival (DMFS), with an adjusted HR of 0.41 (95%CI, 0.25–0.74, P_trend_ = 0.001, 5-year DMFS, 69.0 (57.3–78.1) vs. 50.1 (38.2–60.8), log-rank *p*= 0.012; Figure 4C, lower section of Appendix A). In contrast, in the definitive treatment alone group, an association between folate intake and DMFS was not obvious (adjusted HR comparing high with low folate intake = 0.92, 95%CI, 0.55–1.55, P_trend_ = 0.756, 5-year DMFS 67.3 (55.6–76.6) vs. 54.9 (42.2–65.9), log-rank *p*= 0.354; Figure 4D). This similarity with OS was consistent upon stratification by definitive treatment and cumulative dose of FU. After adjusting for the cumulative dose of FU, a potential difference in the association between folate intake and DMFS by FU-containing IC was observed (Pinteraction = 0.077; Appendix A).

## 4. Discussion

In this study, we found that the association between prediagnostic dietary folate intake and survival statistically significantly differed by regimen in FU-containing IC among patients with locally advanced HNSCC. To our knowledge, this is the first epidemiologic evidence to evaluate the heterogeneity of the survival impact of prediagnostic dietary folate intake using FU-containing chemotherapy. Our finding suggests that chemosensitivity to FU-containing induction chemotherapy inhibiting folate-mediated one-carbon metabolism might be heterogenous and defined by prediagnostic dietary folate intake in the carcinogenic process in HNSCC.

To date, numerous epidemiologic studies have reported evidence for an association between prediagnostic or pre-treatment folate status and the risk of death in cancer patients. Several of these demonstrated that pretreatment folate status was associated with a reduced risk of death among patients with HNSCC, gastric, esophagus, breast, liver, and colorectal cancer [21,37,38,39,41,42,43,44], whereas others studies among patients with ovarian and prostate cancer suggested no clear association [45,46]. Of these, clinical data for FU-containing chemotherapy are limited; Shitara et al. reported that among patients with advanced gastric cancer treated with first-line FU-based chemotherapy, those with high levels of folate intake showed longer overall survival than those with low intake. Further, Kawakita et al. suggested that the survival benefit of higher folate intake was larger in a chemotherapy group than in other treatment groups, albeit without statistical significance [41]. This previous evidence may appear consistent with our present findings. For those cancers with a suggested survival benefit from folate status, as described above [21,37,38,39,41,42,43,44], FU-containing chemotherapy has been recommended as a treatment option [47], and further investigation is accordingly needed.

An association between pretreatment folate intake and the efficacy of FU has been investigated from a range of perspectives. Chardame et al. showed that a stronger response to FU-containing chemotherapy in patients with head and neck cancer was significantly correlated with higher folate pools in their tumor tissues [48]. Given that folate pools in tumor tissue reflect the pretreatment nutritional status of folate intake [49,50,51], it appears plausible that pre-treatment folate intake is associated with a response to FU-containing chemotherapy in head and neck cancer [48]. These previous studies appear to support this interpretation of our present findings.

Allowing that the background mechanism remains unclear, we propose the hypothesis that prediagnostic dietary folate intake might induce a “folate-addict” status which leads to differential sensitivity to FU-containing chemotherapy. An animal study showed that higher folate intake decreased the initiation of carcinogenesis, but that once early tumor lesions were established, the provision of folate increased tumor proliferation [17,18,19,20]. Accordingly, we assume that cancers in patients with high levels of pretreatment folate intake might be “folate-addicted”, possibly with the proliferation of tumor growth resulting from the higher folate intake. The proliferation of this “folate-addict” tumor might depend on the activation of DNA synthesis by folate-mediated one-carbon metabolism [18]. Since FU exerts its antitumor activity through the inhibition of thymidine synthesis (TS) in folate-mediated one-carbon metabolism, the progression of folate-addicted tumors might tend to be suppressed by FU. In addition, folate pools in folate-addict tumor tissue are likely increased with high folate intake [49,50,51], which might in turn result in the formation of ternary complexes with TS and FU and a consequent increase in the antitumor effect of FU [6,8,9].

The direction of evidence to date regarding the association of folate and tumor progression has been inconsistent; preclinical studies and chemoprevention trials indicated that folate fortification increases the proliferation of established tumors [18,19,20,51], whereas several observational epidemiologic studies demonstrated that a higher pretreatment folate status was associated with a lower risk of mortality in several cancers which are potentially treatable with FU, as described above [21,37,38,39,41,42,43,44]. If our hypothesis that “folate-addict” tumors are sensitive to FU, leading to a survival benefit, is true, then the discrepancy above might be partially explained by the interplay of folate-addict tumors and FU. A further direct evaluation of the association between dietary folate intake and FU-containing chemotherapy is warranted.

Of interest, we found that dietary folate intake among FU-containing IC is not related to RFS but is significantly associated with OS and DMFS. Considering evidence from a clinical trial that IC decreased the metastasis of locally advanced HNSCC [22,23,24], it is plausible that FU-containing IC for patients with higher prediagnostic folate might suppress micro-metastasis, thereby improving overall survival.

Our study has several methodological strengths. First, the large majority of patients treated with FU-containing IC had cancers of the oropharynx, hypopharynx, and larynx (80%) at a locally advanced stage, rendering them generally suitable subjects for treatment with FU-containing IC [4,52]. This affirms the external validity of our results. Next, the exposure of interest—dietary folate intake—was measured before treatment, supporting the chronological relation between exposure and outcome. Further, the clinicians who determined treatment in the present study were unaware of exposure status, decreasing the possibility of information bias. Finally, consideration was given to potential confounders. In addition, we also evaluated other nutrients, but only folate intake was significant (Appendix A).

Several methodological limitations also warrant acknowledgement. First, the FFQ was relatively short, and the validity of folate intake was modest. Further, consumption of nutrients from supplements was not quantitively considered in the total vitamin consumption. However, in Japan, supplemental folate use is rare, and a sensitivity analysis among non-supplement-using patients showed the association more clearly. Further, if bias due to the FFQ was present, this would produce non-differential misclassification because exposure to folate intake was the same for both the FU-containing IC followed by definitive treatment and definitive treatment alone groups and would thus bias towards the null. To validate our findings, future studies might consider other methods of evaluating folate, e.g., RBC-folate. Second, because the allocation of FU-containing IC was not carried out at random, a degree of selection bias between the FU-containing IC and definitive treatment alone groups may be present. However, the effect of such misclassification would be limited because of the consistency of our results across most subgroups after stratification by potential confounders. Third, we did not evaluate the use of FU during radiotherapy or adjuvant therapy or during chemotherapy in recurrence. However, any substantial use of FU during definitive treatment or chemotherapy during recurrence might be equivalent in the two groups, thus causing a bias toward the null. Fourth, the exclusion of residual confounding by unevaluated factors, such as human papilloma virus (HPV) infection, cannot be completely ruled out. Fifth, in our investigation, we did not account for the potential confounding effect of folate-metabolizing enzymes. Notably, potential consistent patterns were observed in the subset of patients for whom MTHFR measurements were available. Consequently, future studies that incorporate folate-metabolizing enzymes are warranted to validate and expand upon our findings.

## 5. Conclusions

In conclusion, we showed that the association between prediagnostic folate intake and survival significantly differed by FU-containing IC. Our findings indicate that HNSCC might be heterogenous in terms of sensitivity to folate-mediated one-carbon metabolism, and this might be defined by the heterogenous nutritional condition of folate in the carcinogenic process. Evaluating folate intake preceding the diagnosis of cancer could be pivotal in prognosticating therapeutic outcomes.

## Figures and Tables

**Figure 1 cancers-15-05150-f001:**
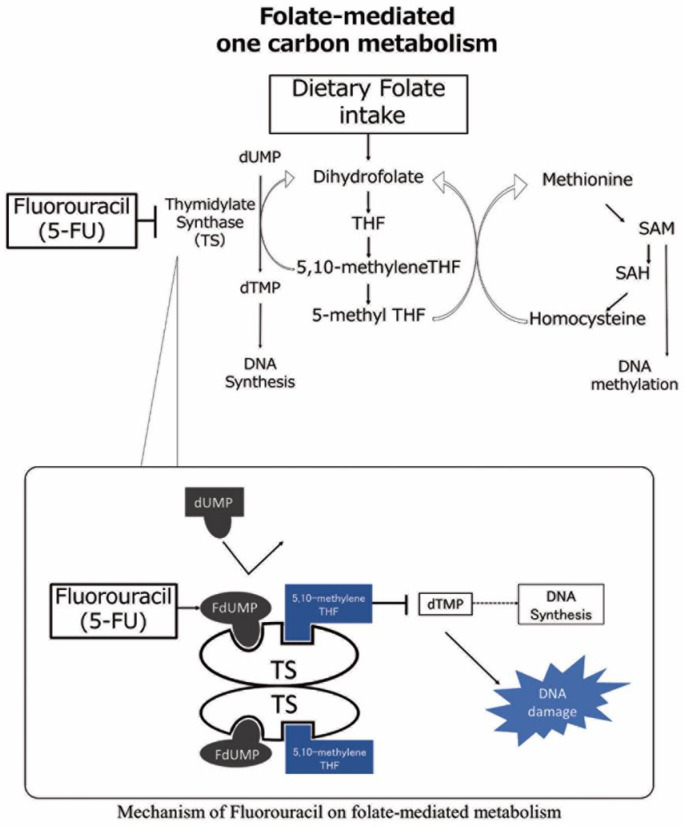
Mechanism of folate-mediated one carbon metabolism and fluorouracil. Dietary folate intake is converted into 5–10 methylene tetrahydrofolate (THF). Then, 5–10 methylene THF and thymidylate synthesis (TS) induce the methylation of dUMP from dTMP, leading to DNA nucleotide synthesis. The 5-fluorouracil (5-FU) active metabolite fluorodeoxyuridine monophosphate (FdUMP) binds to the nucleotide-binding site of TS and forms a stable ternary complex with TS and 5–10 methylene THF, blocking the access of dUMP to the nucleotide-binding site and inhibiting dTMP synthesis, resulting in DNA damage. dTMP; deoxythymidine monophosphate, FdUMP; fluoro-deoxyuridine monophosphate, dUMP; deoxyuridine monophosphate, SAM; S-adenosylmethionine, SAH; S-adenosylhomocysteine.

**Figure 2 cancers-15-05150-f002:**
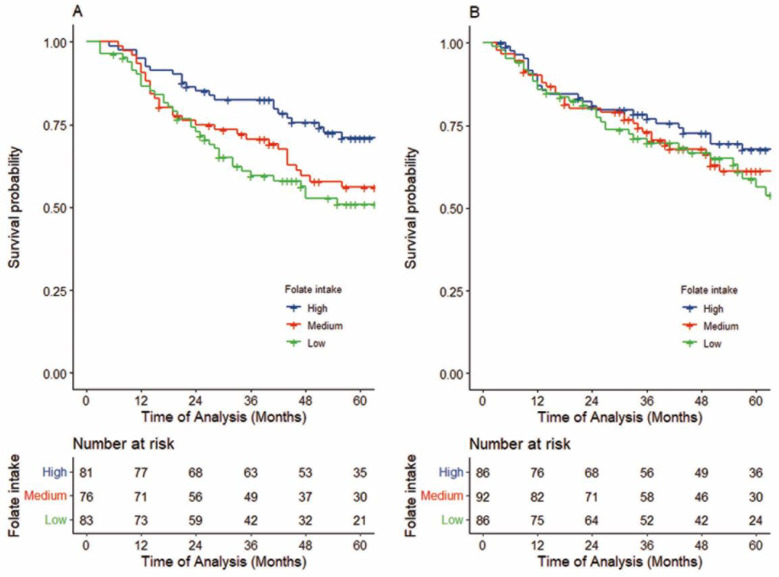
Kaplan–Meier survival curves of OS in patients treated with FU-containing IC followed by definitive treatment (**A**) and by definitive treatment alone (**B**). (**A**) Patients with high folate intake (n = 83) had significantly higher OS than patients with low folate intake (n = 81) (5-year OS: 70.8% vs. 50.8% log-rank *p* =0.020,adjusted HR 0.42, 95%CI, 0.25–0.76, P_trend_ = 0.003). (**B**) Patients with high folate intake (n = 86) did not show better OS than patients with low folate intake (n = 86) (5-year OS: 67.6% vs. 58.6%, log-rank *p* =0.447,adjusted HR 0.83, 95%CI, 0.49–1.42, *p* for trend = 0.491).

**Figure 3 cancers-15-05150-f003:**
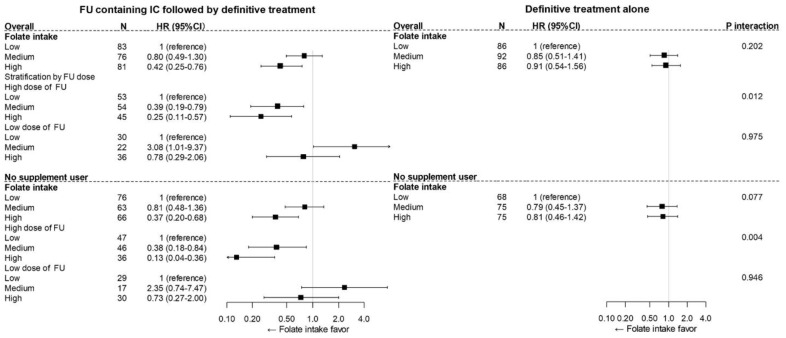
The heterogeneity of the impact of folate intake on OS by FU-containing IC among patients overall (upper) and patients without supplement use (lower). Upper: The difference between the survival impact of folate intake and FU-containing IC did not reached statistical significance (Pinteraction = 0.202); however, the association of folate intake and overall mortality statistically significantly differed between the high-dose FU-containing IC group and the definitive alone group (Pinteraction = 0.012). (lower) Among the patients who were not supplement users, potential heterogeneity in the survival impact of folate between FU-containing IC and definitive treatment alone was revealed (Pinteraction = 0.077), and significant heterogeneity between high-dose FU-containing IC and definitive treatment alone was observed (Pinteraction = 0.004).

**Figure 4 cancers-15-05150-f004:**
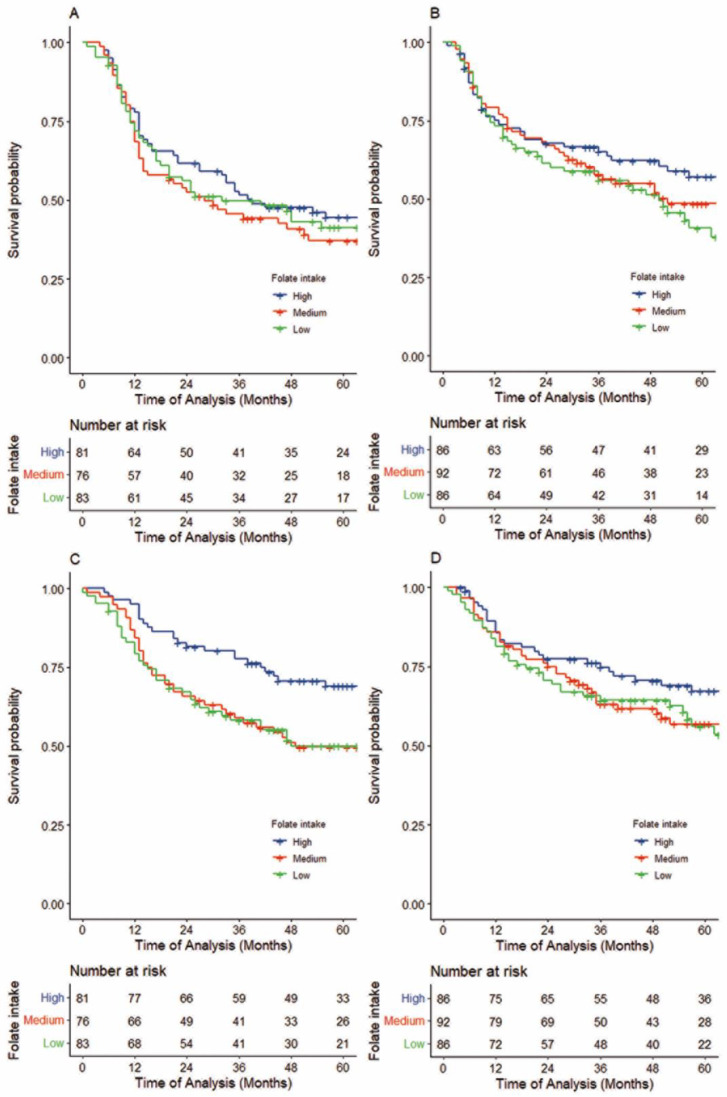
Kaplan–Meier survival curves of recurrence-free survival (RFS) in patients treated with FU-containing IC followed by definitive treatment (**A**), by definitive treatment alone (**B**), and distant metastasis-free survival (DMFS) for patients treated with FU-containing IC followed by definitive treatment (**C**) and by definitive treatment alone (**D**). (**A**,**B**) Folate intake showed no apparent association with RFS in either the FU-containing IC group or the definitive treatment alone group, with adjusted HRs for high compared to low folate intake of 0.88 (95%CI, 0.58–1.36, P_trend_ = 0.596; 5-year-RFS 44.4 (33.0–55.0) vs. 41.2 (30.1–52.2), log-rank *p* = 0.609, Figure 4A) and 0.82 (95%CI, 0.51–1.30, P_trend_ = 0.395; 5-year-RFS 57.0 (45.0–67.3) vs. 40.4 (28.4–52.3) log-rank *p* = 0.351, Figure 4B), respectively. (**C**,**D**) A comparison of high with low folate intake showed that patients with a higher intake in the FU-containing IC group had better DMFS, with an adjusted HR of 0.41 (95%CI, 0.25–0.74, P_trend_ = 0.001; 5-year DMFS, 69.0 (57.3–78.1) vs. 50.1 (38.2–60.8), log-rank *p* = 0.012; Figure 4C). In contrast, with definitive treatment alone, an association between folate intake and DMFS was not obvious (adjusted HR comparing high with low folate intake = 0.92, 95%CI, 0.55–1.55, P_trend_ = 0.756; 5-year DMFS 67.3 (55.6–76.6) vs. 54.9 (42.2–65.9), log-rank *p* = 0.354; Figure 4D).

**Table 1 cancers-15-05150-t001:** Characteristics of patients at baseline.

		Total		FU-Containing IC Followed by Definitive Treatment		Definitive Treatment Alone		
		N = 504	(%)	N = 240	(%)	N = 264	(%)	*p*-Value ^†^
Sex								0.959
	Male	413	(82)	210	(88)	203	(77)	
	Female	91	(18)	30	(13)	61	(23)	
Age (years)								0.200
	<60	214	(42)	107	(45)	107	(41)	
	≥60	290	(58)	123	(51)	167	(63)	
ECOG PS								0.016
	0	293	(58)	153	(64)	140	(53)	
	1	191	(38)	82	(34)	109	(41)	
	2	20	(4)	5	(2)	15	(6)	
Primary site								0.016
	Oral cavity	201	(40)	52	(22)	149	(56)	
	Oropharynx	114	(23)	85	(35)	29	(11)	
	Hypopharynx	136	(27)	85	(35)	51	(19)	
	Larynx	53	(11)	18	(8)	35	(13)	
UICC stage								0.004
	III	142	(28)	53	(22)	89	(34)	
	IV	362	(72)	187	(78)	175	(66)	
UICC T classification								0.132
	1	33	(7)	13	(5)	20	(8)	
	2	165	(33)	87	(36)	78	(30)	
	3	153	(30)	78	(33)	75	(28)	
	4	152	(30)	62	(26)	90	(34)	
	x	1	(0)	0	(0)	1	(0)	
UICC N classification								<0.001
	0	104	(21)	38	(16)	66	(25)	
	1	111	(22)	38	(16)	73	(28)	
	2	268	(53)	147	(61)	121	(46)	
	3	21	(4)	17	(7)	4	(2)	
Definitive treatment								0.013
	Surgery	223	(44)	97	(40)	126	(48)	
	Radiotherapy	281	(56)	143	(60)	138	(52)	
Cumulative smoking								0.001
	Non-smoker	111	(22)	30	(12)	81	(31)	
	Light (<20PY)	71	(14)	37	(15)	34	(13)	
	Moderate (20PY to <30PY)	127	(25)	60	(23)	67	(25)	
	Heavy (≥30PY)	181	(36)	104	(43)	77	(30)	
	Unknown	14	(3)	9	(4)	5	(2)	
Alcohol consumption								0.001
	Non-drinker	115	(23)	40	(17)	75	(28)	
	Light	112	(22)	45	(19)	67	(25)	
	Moderate	107	(21)	56	(23)	51	(19)	
	Heavy	167	(33)	98	(41)	69	(26)	
	Unknown	3	(1)	1	(0)	2	(1)	
Folate intake ^‡^								0.746
	Low	169	(34)	83	(35)	86	(33)	
	Medium	168	(33)	76	(32)	92	(35)	
	High	167	(33)	81	(34)	86	(33)	
Vitamin supplement use								0.094
	No	423	(84)	205	(85)	218	(83)	
	Yes	69	(14)	33	(14)	36	(14)	
	Unknown	12	(2)	2	(1)	10	(4)	
Cumulative dose of FU during IC term								
	High-dose ^§^	-		152	(63)	-		
	Low-dose ^§^	-		88	(37)	-		

ECOG PS, Eastern Cooperative Oncology Group performance status; UICC, International Union Against Cancer. ^‡^ Low (≤237 μg/d: HERPACC2 and ≤288 μg/dHERPACC3; lowest tertile), medium (>237 and ≤322 μg/d and >288 and ≤380 μg/d; middle tertile), and high (>322 μg/d and >380 μg/d; highest tertile). ^§^ High-dose: the cumulative dose of FU during IC was more than 8000 mg/m^2^, equivalent to two cycles of FU + CDDP (3 week). Low-dose: the cumulative dose of FU during IC was less than 8000 mg/m^2^ in terms of IC. ^†^ Chi-squared test or Fisher’s exact test.

**Table 2 cancers-15-05150-t002:** Impact of folate intake on overall survival.

	FU-Based IC Followed by Definitive Treatment							Definitive Treatment Alone							
Folate Intake ^†^	N (%)	Event	5-Year OS (%) (95%CI)	Crude HR (95%CI)	*p*-Value	Adjusted ^‡^ HR (95%CI)	*p*-Value	N (%)	Event	5-Year OS (%) (95%CI)	Crude HR (95%CI)	*p*-Value	Adjusted ^‡^ HR (95%CI)	*p*-Value	*p* for Interaction
Overall	N = 240							N = 264							0.202 ^§^
Low	83 (35)	38	50.8 (37.6–61.6)	1 (reference)		1 (reference)		86 (33)	36	58.6 (45.8–69.2)	1 (reference)		1 (reference)		
Medium	76 (32)	34	56.0 (43.6–66.7)	0.86 (0.54–1.37)	0.516	0.80 (0.49–1.30)	0.365	92 (35)	34	61.0 (49.1–71.0)	0.85 (0.53–1.36)	0.504	0.82 (0.49–1.37)	0.449	
High	81 (34)	24	70.8 (59.0–80.6)	0.50 (0.30–0.84)	0.009	0.42 (0.25–0.76)	0.003	86 (33)	29	67.6 (55.7–77.0)	0.75 (0.46–1.22)	0.245	0.83 (0.49–1.42)	0.504	
				trend *p* = 0.009		trend *p* = 0.004					trend *p* = 0.243		trend *p* = 0.491		
Stratification by cumulative dose of FU during IC terms															
High cumulative dose of FU during IC term in FU-containing IC (N = 152) ^§§^															0.012 ^¶^
Low	53 (35)	26	48.0 (32.8–61.7)	1 (reference)		1 (reference)									
Medium	54 (36)	18	67.2 (51.8–78.7)	0.55 (0.30–1.01)	0.053	0.35 (0.17–0.74)	0.006								
High	45 (30)	9	79.7 (63.1–89.4)	0.29 (0.13–0.62)	0.002	0.22 (0.10–0.53)	0.001								
				trend *p* = 0.001		trend *p* = <0.001									
Low cumulative dose of FU during IC term (N = 88) ^§§^															0.975 ^†^
Low	30 (34)	12	56.1 (35.5–72.5)	1 (reference)		1 (reference)									
Medium	22 (25)	16	29.6 (12.2–49.3)	1.85 (0.87–3.93)	0.108	3.18 (1.04–9.71)	0.042								
High	36 (41)	15	59.7 (41.5–73.9)	0.90 (0.87–3.93)	0.784	0.78 (0.29–2.05)	0.608								
				trend *p* = 0.683		trend *p* = 0.305									
Stratification by definitive treatment															
Surgery N = 97								N = 126							0.762 ^§^
Low	27 (28)	11	54.2 (31.9–72.0)	1 (reference)		1 (reference)		35 (28)	14	63.9 (43.2–78.8)	1 (reference)		1 (reference)		
Medium	37 (38)	21	45.6 (28.6–61.1)	1.37 (0.66–2.86)	0.394	1.05 (0.44–2.46)	0.919	51 (40)	17	60.1 (42.5–73.9)	0.93 (0.45–1.94)	0.849	1.16 (0.50–2.67)	0.733	
High	33 (34)	9	70.1 (50.0–83.4)	0.55 (0.23–1.35)	0.199	0.35 (0.13–0.93)	0.034	40 (32)	11	72.1 (54.1–84.1)	0.59 (0.26–1.34)	0.214	0.89 (0.33–2.38)	0.823	
				trend *p* = 0.215		trend *p* = 0.025					trend *p* = 0.215		trend *p* = 0.854		
Stratification by cumulative dose of FU during IC terms															
High cumulative dose of FU during IC term in FU-containing IC (N = 45) ^§§^															0.031 ^¶^
Low	10 (22)	7	25.0 (4.0–54.8)	1 (reference)		1 (reference)									
Medium	22 (49)	10	56.3(32.6–74.5)	0.45(0.16–1.22)	0.118	0.12 (0.19–0.74)	0.023								
High	13 (29)	1	88.9 (43.3–98.3)	0.06 (0.07–0.51)	0.010	0.03 (0.00–0.38)	0.007								
				trend *p* = 0.002		trend *p* = 0.005									
Low cumulative dose of FU during IC term (N = 52) ^§§^															0.361 ^†^
Low	17 (37)	4	75.3 (45.7–89.9)	1 (reference)		1 (reference)									
Medium	15 (33)	11	30.0 (9.5–54.0)	3.65 (1.15–11.50)	0.027	2.54 (0.48–13.15)	0.268								
High	20 (43)	8	58.0(33.0–76.5)	1.62 (0.48–5.38)	0.784	0.25 (0.05–1.40)	0.115								
				trend *p*= 0.593		trend *p* = 0.64									
Radiotherapy		N =143						N = 138							0.302 ^§^
Low	56 (39)	27	48.9 (34.262.1)	1 (reference)		1 (reference)		51 (37)	22	54.7 (38.0–68.7)	1 (reference)		1 (reference)		
Medium	39 (27)	13	67.1 (49.1–89.1)	0.54 (0.25–1.07)	0.078	0.49 (0.23–1.05)	0.067	41 (30)	17	62.5 (45.5–75.5)	0.86 (0.45–2.62)		0.70 (0.34–1.42)	0.324	
High	48 (34)	15	71.6 (56.0–82.5)	0.47 (0.25–0.91)	0.026	0.56 (0.26–1.17)	0.121	46 (33)	18	63.3 (46.6–76.4)	0.82 (0.44–1.54)		0.70 (0.35–1.42)	0.327	
				trend *p* = 0.023		trend *p* = 0.083					trend *p* = 0.551		trend *p* = 0.319		
Stratification by cumulative dose of FU during IC terms															
High cumulative dose of FU during IC term in FU-containing IC (N = 107) ^§§^															0.142 ^¶^
Low	43 (40)	19	53.5 (36.2–68.2)	1 (reference)		1 (reference)									
Medium	32 (30)	8	76.5 (56.4–88.2)	0.47 (0.19–1.11)	0.085	0.30 (0.10–0.88)	0.028								
High	32 (30)	8	76.5 (56.7–88.1)	0.42 (0.17–1.01)	0.052	0.30 (0.10–0.91)	0.033								
				trend *p*= 0.042		trend *p* = 0.022									
Low cumulative dose of FU during IC term (N = 36) ^§§^															0.764 ^†^
Low	13 (36)	8	33.8 (10.5–59.4)	1 (reference)		1 (reference)									
Medium	7 (19)	5	28.6 (4.1–61.2)	0.92 (0.30–2.82)	0.879	2.46 (0.16–37.81)	0.518								
High	16 (44)	7	61.9 (33.9–80.8)	0.55 (0.19–1.54)	0.258	0.74 (0.10–5.20)	0.761								
				trend *p* = 0.253		trend *p* = 0.651									

^‡^ adjusted by sex, age, performance status, smoking, alcohol consumption, primary site, definitive therapy, UICC T classification, UICC N classification, energy, supplement use. ^§^ Interaction between FU-based IC and definitive treatment alone; ^¶^ interaction between high FU-based IC and definitive treatment alone; ^†^ interaction between high and low cumulative doses of FU during IC term; *p* for interaction between high cumulative dose of FU during IC term and definitive treatment alone”. ^§§^ High: Cumulative dose of FU during IC terms was more than 8000 mg/m^2^, equivalent to 2 cycles. Low: cumulative dose of FU during IC terms was less than 8000 mg/m^2^ in terms of IC.

## Data Availability

Data sharing not applicable.

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
