# Peer review of "The Heterogeneous Impact of Prediagnostic Folate Intake for Fluorouracil-Containing Induction Chemotherapy for Head and Neck Cancer"

_cancers, 2023, doi:10.3390/cancers15215150_

Round 1

Reviewer 1 Report

This is an important study for the treatment of OSCC.
Statistically everything is in order. In other respects, this is also a well thought out and carefully conducted study.
I have only a few suggestions for improvement:
1) Is there a reference for figure 1? Where did this information come from? Please indicate the abbreviations!
2) Wherever it says Ptrend, the term "trend" belongs as an index.
3) The Kaplan-Meier curves are graphically unclear, especially the numerous vertical lines are disturbing. Could this be optimized? Stata may not be the best choice for this. Maybe with RStudio or with GraphPad-Prism? Or Sigma plot?
4) Have log-rank tests been done?
5) What is the clinical implication of this study? Should dietary supplementation of folic acid be included in the treatment plan?

Reviewer 2 Report

Brief summary 

The paper focuses the attention on a very interesting topic, as the correlation between folate intake and survival in head and neck is not widely studied, especially considering neoadjuvant setting.  

Moreover, it is a original production.  

This is a clear paper 

The conclusions are interesting and add advances in the current scientific knowledge.  

No ethical problems are found in this study  

I would like to make some suggestions and I have few questions 

General concept comments 

Abstract 

Please specify what “definitive treatment” is, as it can be surgical excision of the tumor with free margins, excision of the tumor+ neck lymph node dissection etc.  

Introduction 

I suggest to expand more the head and neck squamous cell carcinoma part of the introduction  

Methods and results 

Considering the methos bias I would suggest reconsidering this part maybe by adding other variable such as genetic polymorphism 

english level is good 

Round 2

Reviewer 2 Report

The paper is better in this revised version

Good quality